# Transcriptome analysis of Catarina scallop (*Argopecten ventricosus*) juveniles treated with highly-diluted immunomodulatory compounds reveals activation of non-self-recognition system

Jesús Antonio López-Carvallo[1], José Manuel Mazón-Suástegui[1], Miguel Ángel Hernández-Oñate[2]*, Dariel Tovar-Ramírez[3], Fernando Abasolo-Pacheco[4], Rosa María Morelos-Castro[5], Guadalupe Fabiola Arcos-Ortega[5]*

1 Laboratorio Experimental de Cultivo de Moluscos, Centro de Investigaciones Biológicas del Noroeste, La Paz, México, 2 CONACyT, Centro de Investigación en Alimentación y Desarrollo A.C, Hermosillo, Sonora, México, 3 Laboratorio de Fisiología Comparada y Genómica Funcional, Centro de Investigaciones Biológicas del Noroeste, La Paz, México, 4 Facultad de Ciencias Agrarias, Universidad Técnica Estatal de Quevedo, Quevedo, Los Ríos, Ecuador, 5 Laboratorio de Imunogenómica Marina, Centro de Investigaciones Biológicas del Noroeste, La Paz, México

* farcos04@cibnor.mx (GFAO); miguel.hernandez@ciad.mx (MAHO)

## Abstract

Marine bivalve hatchery productivity is continuously challenged by apparition and propagation of new diseases, mainly those related to vibriosis. Disinfectants and antibiotics are frequently overused to prevent pathogen presence, generating a potential negative impact on the environment. Recently, the use of highly diluted compounds with immunostimulant properties in marine organisms has been trailed successfully to activate the self-protection mechanisms of marine bivalves. Despite their potential as immunostimulants, little is known about their way of action. To understand their effect, a comparative transcriptomic analysis was performed with *Argopecten ventricosus* juveniles. The experimental design consisted of four treatments formulated from pathogenic *Vibrio* lysates at two dilutions: [(T1) *Vibrio parahaemolyticus* and *Vibrio alginolyticus* 1D; (T2) *V. parahaemolyticus* and *V. alginolyticus* 7C]; minerals [(T3) PhA+SiT 7C], scorpion venom [(T4) ViT 31C]; and one control (C1) hydro-alcoholic solution (ethanol 1%). The RNA sequencing (RNAseq) analysis showed a higher modulation of differentially expressed genes (DEG) in mantle tissue compared to gill tissue. The scallops that showed a higher number of DEG related to immune response in mantle tissue corresponded to T1 (*V. parahaemolyticus* and *V. alginolyticus* lysate) and T3 (Silicea terra® - Phosphoric acid®). The transcriptome analysis allowed understanding some interactions between *A. ventricosus* juveniles and highly-diluted treatments.

**Data Availability Statement:** The transcriptome raw reads were deposited in the Gene Expression Omnibus of NCBI under accession number PRJNA596225. All other relevant data are within the manuscript and its Supporting Information files.

**Funding:** This study was funded by the Sectoral Fund for Research for Education of México; projects Basic Science CONACyT 258282 and PROINNOVA-CONACyT 24177 under the academic responsibility of JMMS. JALC is the recipient of a doctorate fellowship (CONACYT-301921), under the academic direction of GFAO and JMMS.

**Competing interests:** The authors have declared that no competing interests exist.

# Introduction

Marine bivalve hatchery production has been challenged by apparition and propagation of new diseases [1], mainly those related with *Vibrio* spp. [2,3], leading to high mortality and economical losses. Recent studies have suggested that proliferation of pathogenic vibrio species will increase in a near future due to sea surface water warming [4,5], and most of the time, hatchery conditions have been favourable to *Vibrio* spp. [6]. Traditionally, hatcheries around the world have routinely implemented the prophylactic application of antibiotics and disinfectants to prevent proliferation or kill pathogenic bacteria, as standard prophylactic or therapeutic procedures [7,8].

Chemotherapeutic applications have been restricted by governmental institutions [2] for their undesirable effects, such as selection of antibiotic-resistant bacteria [7,8], their implications in dissolution of mercury compounds when contaminated water is in contact with marine sediment [9] and alterations in the beneficial gastrointestinal microbiota of cultured organisms [10].

Immunostimulants have been proposed as promising and sustainable alternatives to avoid massive bivalve mortalities in hatchery and reduce or replace the use of antibiotics [11]. Some immunomodulatory compounds can prophylactically activate the organisms' self-protection instead of killing the pathogen by chemical compounds, which may compromise the organisms' health and environment. Within the immunostimulant category, highly-diluted immunomodulatory compounds (HDIC) formulated from bacterial lysates, scorpion venom, silica and phosphoric acid have been recently suggested as eco-friendly and sustainable options to activate the immune system without affecting the general condition index of marine organisms [12]. HDIC are inexpensive to produce because serial dilutions at decimal (D, 1:10) or centesimal (C, 1:100) scale [13] are used, and the action mode of these treatments depends on their dilution grade. Low decimal dilution (D) effects are attributed to diluted molecules and nanoparticles while high centesimal dilution (C) effects are attributed to nanoparticles and electromagnetic fields [14], which are sensed by traditional molecular receptors or photoelectrochemical sensing system activated by ultra-weak photo emission signals [15]. These highly-diluted molecules work as low intensity danger signals that generate endogenous amplification by hormesis effect, time-dependent sensitisation and stochastic resonance process [16,17].

Low concentrations of attenuated bacteria [18,19] have already proven to activate immune response in different organisms and highly-diluted concentrations of bacterial lysates, Silicea terra® (Similia®, CDMX, MX), Phosphoric acid® (Similia®, CDMX, MX) and Vidatox® (Labiofam®, Habana, Cuba) have been successfully used in marine organisms to improve response against pathogens [12]. Previous studies have demonstrated that survival of *Argopecten ventricosus* juveniles increased when the organisms were previously therapeutically treated by Silicea terra® and Phosphoric acid® 31C, and then challenged against highly pathogenic bacteria, such as *Vibrio parahaemolyticus* [20]. Furthermore, increase in growth, enzymatic antioxidant activity, and haemocyte proliferation were reported when juvenile scallops were treated prophylactically with diluted and highly-diluted pathogenic bacterial lysates (1D and 7C) for a 21-day period [21]. Although those immunomodulatory compounds are known to allow organisms to activate immune response and increase survival when they face pathogen infections, little is known about how they work on the immune response of marine organisms.

Transcriptomic (RNA-seq) analysis allows discovering potential and novel action mechanisms. Particularly, these analyses have provided information to analyse complex relations among disease, drugs, and organisms [22]. Moreover, RNA-seq has been used in marine bivalves to understand physiological stress [23,24], toxicological effects [25,26], and resistance

to disease and immunology [27,28]. In this sense, the RNA-seq analysis helps to understand the molecular response activated by the effect of time-dependent diluted immunomodulatory treatments, which allowed identifying the down- and up-regulated genes of juvenile scallops in response to HDIC.

Because the immune defense mechanisms in response to HDIC formulated by bacterial lysates, Silicea terraⓇ, Phosphoric acidⓇ and VidatoxⓇ remain unknown, this study performed a comparative analysis of the mantle and gill transcriptome profile treated with those HDIC. These data will provide important information about the genes and mechanisms that are being regulated in *A. ventricosus* scallop and contribute to understanding how HDIC acts on marine organisms' response.

This study also provided for the first time whole transcriptome data of the Catarina scallop *A. ventricosus* juveniles, which was selected as a model organism because of its importance as fishing resource in Baja California Sur, Mexico. Its wild populations have been decreasing throughout the years [29], and spat production at hatchery level faces high mortality events as it is a species highly susceptible to *vibrosis* [30], one of the main bacteria present in hatchery culture water.

## Materials and methods

### Scallop acquisition and experimental design

The non-governmental association Noroeste Sustentable (NOS) provided 1500 juvenile scallops (average length 1.98 ± 0.1 cm) from Bahia de La Paz, Mexico to perform the experiment. Scallops were acclimated for one week in a nursery upwelling recirculating system with constant food concentration of 150 000 cell mL$^{-1}$ (*Isochrysis galbana; Chaetoceros calcitrans*; cellular proportion 1:1). Filtered seawater (1 μm, activated carbon and ultraviolet (UV) irradiation) at 24°C and 38.5 ± 0.5 UPS salinity were continuously supplied to allow water change totally every day.

After acclimatization, scallops were transferred to 15 experimental units (36-L container) with 52 scallops each (three experimental units per treatment). During the experimental period (21 days), scallops were kept in an open-recirculating flow system that provided filtered and treated seawater (1 μm, activated carbon and UV irradiation) with a blend of *I. galbana* and *C. calcitrans* microalgae (199 607 794 cell/organism/day) at 23.5 ± 0.5°C, and 38.5 ± 0.5 UPS salinity. A 21-day experimental period was sufficient to strengthen the immune system of this species with HDIC-based treatments [20].

Four experimental HDIC treatments [T1: *V. parahaemolyticus* and *V. alginolyticus* lysate diluted at 1:10 (1D); T2: *V. parahaemolyticus* and *V. alginolyticus* lysate diluted at 1: $10^{-14}$ (7C); T3: Silicea terraⓇ (SimiliaⓇ, Farmacia Homeopática NacionalⓇ, CDMX, MX) and Phosphoric acidⓇ (SimiliaⓇ, Farmacia Homeopática NacionalⓇ, CDMX, MX) diluted at $1X10^{-14}$ (7C); T4: VidatoxⓇ (LabiofamⓇ, Habana, Cuba) diluted at $1X10^{-62}$ (31C)], and one control treatment [C1: hydro-alcoholic solution (1%)] were assayed by triplicate under laboratory conditions. All HDIC or control treatments were supplied to experimental units as used in our previous studies [20,21]. The open water and food flow was cut three hours a day to favour treatment uptake through the mantle and gill tissues. All treatments were added in liquid form directly to seawater in each experimental unit (100 μl L$^{-1}$). Treatments T1, T2 and T3, which consisted of two HDIC each, were provided alternately.

At the end of the experiment, organisms for each experimental condition were taken. The mantle and gill tissues of three sampled scallops of each experimental replicate unit were excised, separately fixed in RNAlater solution (#AM7020, ThermoFisher, Scientific, Waltham, MA, USA) and stored at -80°C for analysis. Mantle and gill tissue were selected as target

organs because they have been highly implicated with immune response and have been related with pattern recognition receptors in marine bivalves [31,32].

## Highly-diluted immunomodulatory compounds (HDIC) formulation

All treatments and control were made following the methodology previously reported for HDIC preparation [21,33], which is described below.

T1 (ViP 1D + ViA 1D) and T2 (ViP 7C + ViA 7C) were developed at Centro de Investigaciones Biológicas del Noroeste (CIBNOR), La Paz, Baja California Sur, Mexico and formulated in decimal (T1) or centesimal (T2) dilution/succussion (agitated) from a concentrate of pathogenic strains of *V. parahaemolyticus* (CAIM 170; www.ciad.mx/caim » ViP) and *V. alginolyticus* (CAIM 57; www.ciad.mx/caim » ViA). These strains were isolated from high mortality events in marine bivalves [34]. For each strain, a concentrated solution was prepared from wet biomass obtained by centrifugation (3 300 g, 4°C, 20 min) of 2 L of *Vibrio* culture ($105 \times 10^{-6}$ CFU mL-1) in marine broth 2216 (BD Difco™, USA). The wet biomass of each strain (15 mL) was fully inactivated by three freeze-unfreeze cycles of −80 and 24°C; the strain biomass was vortexed (Benchmark mixer™, Benchmark Scientific Inc. Sayreville, NJ, USA) between each cycle for 2 min (3 200 rpm). The inactivated product was topped up to the original culture volume (2 L) using ethanol 87° (Similia® purchased at Farmacia Homeopática Nacional®, CDMX, MX) and vortexed (Benchmark mixer™, Benchmark Scientific Inc. Sayreville, NJ, USA) for 2 min (3 200 rpm) to get a final concentrated solution; with this concentrate, decimal (D, 1:10) and centesimal (C, 1:100) dilutions were prepared by a serial successive dilution/succussion process (3200 rpm, 2 min, Benchmark mixer™, Benchmark Scientific Inc. Sayreville, NJ, USA) until a working solution 1D ($1X10$) and 7C ($1X10^{14}$) had been reached.

Treatments T3 (PhA 7C + SiT 7C) and T4 (ViD 31C) were made from commercial drugs diluted in ethanol 87°. T3 consisted of centesimal (C) preparations (1:100 dilution/succussion) of the commercial drugs Phosphoric acid® 6C (Similia®, CDMX, MX; dilution $1 \times 10^{12}$) and Silicea terra® 6C (Similia®, CDMX, MX; dilution $1 \times 10^{12}$) while T4 (ViD 31C) consisted of a centesimal dilution (1:100 dilution/succussion) of the commercial drug Vidatox® 30C (Labiofam®, Habana, Cuba; dilution $1 \times 10^{60}$) made from blue scorpion *Rhopalurus junceus* venom. To prevent potential ethanol side effects, distilled water was used as a final dilution vehicle for all HDIC working solutions.

## RNA extraction and cDNA synthesis

To extract RNA, the tissue (mantle and gill) samples from three scallops (~100 mg) were collected and pooled. Each pool was considered as the biological sample. For the Real-Time quantitative polymerase chain reaction (RT-qPCR) analysis, three pools of each tissue were used per treatment (Fig 1A). For transcriptomic analysis, two pools (~100 mg each) were used per treatment (Fig 1A). The RNA extraction was assessed following the methodology previously described for the scallop *Nodipecten subnodosus* RNA extraction [35], using the TriPure reagent (Roche Diagnostics, Indianapolis, IN, USA), followed by ethanol/chloroform purification and DNAse cleaning. Samples were determined for RNA concentration using a NanoDrop 2000/2000c® spectrophotometer (Thermo Fisher Scientific, Waltham, MA, USA), and electrophoresis was performed to evaluate RNA quality. Good quality RNA from a total of 20 Pools (two per treatment per tissue) were sent to the Laboratorio Nacional de Genómica para la Biodiversidad (LANGEBIO, Laboratory of Genomics Services, CINVESTAV, Campus Irapuato, GTO, MX) to create and sequence the RNA-seq libraries using the Illumina Next-Seq® (San Diego, CA, USA). For RT-qPCR analysis, the cDNA synthesis was performed following the methodology previously reported to quantify gene relative expression levels in *N. subnodosus* [35].

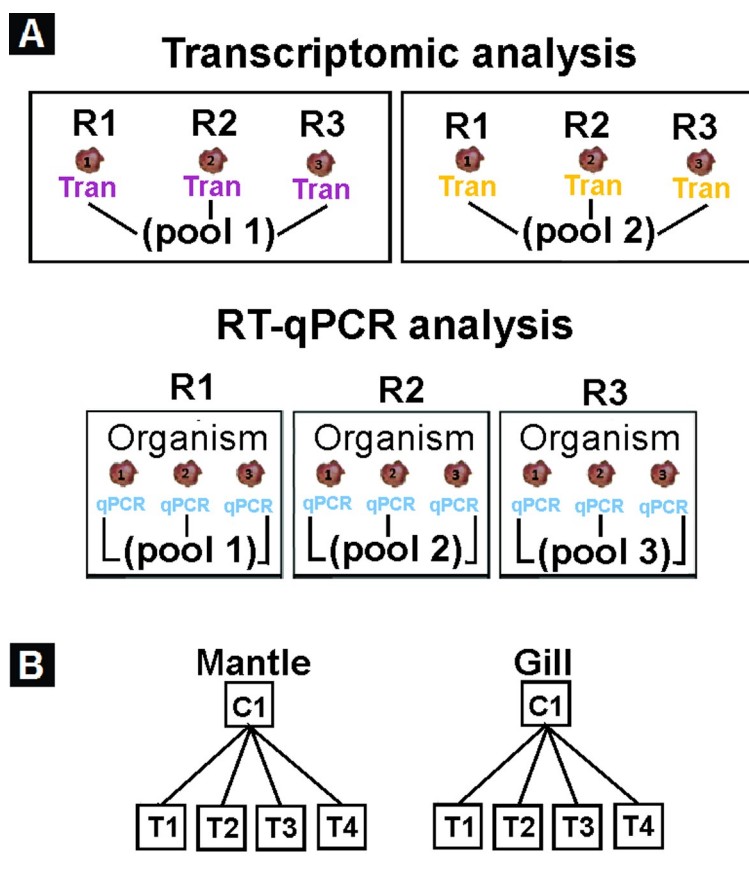

**Fig 1. Sampling collection and data analysis strategy.** (A) Work diagram for sample pooling. Each diagram represents the pooling strategy for each experimental condition. Purple and yellow colours denote organisms used for transcriptome pools (three organisms per replicate), and blue denotes organisms used for qPCR pools (three organisms per replicate); (B) RNAseq analysis strategy. For each tissue, treatments were separately compared against the control. T1: *V. parahaemolyticus* and *V. alginolyticus* lysate 1D; T2: *V. parahaemolyticus* and *V. alginolyticus* lysate 7C; T3: Silicea terra® (Similia®, CDMX, MX) and Phosphoric acid® (Similia®, CDMX, MX) 7C; T4: Vidatox® (Labiofam®, Habana, Cuba) 31C, and control treatment C1: hydro-alcoholic solution (1%).

## Illumina sequencing

For RNA-seq library preparation, the two RNA pools were selected from mantle and gill of each experimental condition. A total of 20 libraries were prepared using the RNA extractions. In addition, 12 libraries were prepared with the organisms treated with HDIC formulated by sodium metasilicate—Phosphoric acid 1D plus two controls (C2: diluted ethanol 1:100 and C3: nothing added) to increase assembly and annotation quality. Libraries were validated by 2100 Bioanalyzer Instrument (Agilent, Santa Clara, CA, USA) and sequenced with Illumina NextSeq® (San Diego, CA, USA) platform on mode paired-end (2 X 150 high). Generation of libraries and sequencing were performed by LANGEBIO (Laboratory of genomics services, CINVESTAV, Campus Irapuato, GTO, MX).

## Transcriptome, annotation and statistical analyses

The raw reads obtained by the Illumina NextSeq® platform (San Diego, CA, USA) were first analysed with the FastQC software [36] to determine quality sequences; then, the reads were filtered using the Trimmomatic program [37] by removing low-quality reads (Q < 25),

ambiguity ("N"), reads under 50 bp and trimming the adapters. The high-quality filtered reads were used to perform the *de novo* transcriptome assembly with Trinity software [38] with the "non-normalized reads" option. This process was performed in a WorkStation with 10 cores and 512 GB RAM. Once the assembly process was completed, the quality and integrity of the transcriptome were analysed. The high-quality filtered reads were mapped to assembled transcripts with RSEM software which used Bowtie2 for the alignment [39], and the expression values were normalised to transcripts per million (TPM) for each transcript. Those with low expression values (TPM < 0.01) were removed. The assembled transcripts were compared against the non-redundant nucleotide and protein databases of the National Center for Biotechnology Information (NCBI) using BLAST with an e-value $< 1e^{-5}$, sequences with hit to viruses, bacteria, fungus, and so on (considered contaminant sequences) were removed using the SeqClean tool [40]. Transcriptome redundancy was first removed by CD-Hit using 97% of identity as clustering threshold [41] and then by iAssembler program [42]. The statistical, quality and completeness analyses of the assembled transcriptome were performed using the PRINSEQ [43], TransRate [44], and BUSCO [45] tools.

To determine the functional annotation based on homology, the filtered transcriptome was compared against the public protein non-redundant nucleotides (NCBI), Swiss-Prot and RefSeq databases using BLASTX (e-value $< 1e^{-5}$). In addition, a BLASTX (e-value $< 1e^{-5}$) against predicted proteins of the *Crassosrea gigas*, *Crassostrea virginica*, *Mytilus galloprovincialis* and *Mizuhopecten yessoensis* genomes was performed. The functional annotation was assigned according to the BLAST hits. Finally, the functional categorisation was designated based on the comparison with the Gene Ontology (GO), InterPro and Kyoto Encyclopaedia of Genes (KEGG) databases using the BLAST2GO software [46]. The raw reads were deposited in the Gene Expression Omnibus of NCBI under accession number PRJNA596225.

## Differential expression and enrichment analyses

Only were 20 libraries corresponding to the treatments (T1, T2, T3, T4) and control (C1) considered for the posterior analysis. High-quality alignment of the filtered reads to the filtered transcriptome and quantification of reads were carried out using the RSEM software [39]. Raw counts of each transcript of each library were used to generate a matrix with all the experimental conditions and replicates. These raw counts were normalised to fragments per kilobase per million mapped reads (FPKM). To determine dispersion between samples and biological replicates, the Pearson correlation value was calculated between all experimental libraries. Only were libraries with Pearson's correlation > 0.88 between replicates kept for subsequent analyses.

Once every biological replicate was validated, transcripts lowly expressed were removed and only transcripts with a FPKM > 0.3 in at least two replicates of at least one treatment were kept, as edgeR reduce analysis quality when very low expressed genes are use [47]. Differentially expressed genes in Catarina scallop juveniles treated with the HDIC (Fig 1B) versus control without treatment were identified with the edgeR package [47], using a statistical method based on the generalised linear model (GLM). To estimate the variance between samples, tagwise dispersion was calculated. Transcripts with FDR < 0.01 and Log$_2$FoldChange >1 (Log$_2$FC) were considered differentially expressed. DEG transcripts were grouped by tissue as down- and up-regulated for each treatment, and then Venn diagrams were generated including all the comparisons between DEG of treatments.

Gene Ontology and KEGG category enrichment analyses were performed using BLAST2GO [46] and KOBAS [48], respectively, with FDR < 0.05. The function to the most specific terms was used to reduce the result-set of over-represented GO terms. Heatmaps and hierarchical clustering were performed using the gplots and hclust in R [49].

## RT-qPCR validation

To validate differential expression from RNAseq data, 10 target genes ($Log_2FC > 1.5$ and $FDR < 0.01$) and four constitutive genes (CV% < 0.3, $Log_2FC$ between -1.5 and +1.5 and FPKM > 0.3) from the transcriptome were selected (S1 Table). Primers were designed using the software Primer3 [50] and then analysed with OligoAnalyzer Tool [51]. The most stable of the constitutive genes for each tissue was selected using RefFinder software [52] to express relative expression. Primers were evaluated and target sequences validated by sequencing analysis (Macrogen, Seoul, Korea).

For the RT-qPCR analysis, three biological replicates were assessed per treatment (each biological replicate was performed by triplicate) using previously synthesised cDNA from mantle and gill. RT-qPCR analysis was performed as previously reported to quantify gene relative expression levels in *N. subnodosus* [35] following Minimum Information for Publication of Quantitative Real-Time PCR Experiments (MIQE) guidelines [53]. Percentage of efficiency of each primer pair was obtained from efficiency curves from six serial dilutions (1:5) [53]. Reactions of RT-qPCR were performed with the equipment using 2X Eva-Green® (ThermoFisher, Scientific, Waltham, MA, USA) PCR mix. Reaction mixtures were made in 15 μL, including 10 μL mix (0.45 U of GoTaq Flexi DNA polymerase (Promega, Madison, WI, USA), 2.5 mM MgCl2, 1x Go Taq Flexi Buffer, 0.2 mM dNTP Mix (Promega, Madison, WI, USA), 2x EvaGreen fluorescent dye (Biotium, Inc. Fremont, CA, USA), 0.15–0.45 μM each primer) and 5μL cDNA (diluted to 50 ng μL$^{-1}$). The amplification parameters were 4 min at 94˚C, followed by 39 cycles of 9 s at 95˚C, 30 s at 60˚C and 30 s at 72˚C; finally, a melt curve was added to confirm amplification of specific products. Data were analysed using Ct values and the $2^{-\Delta\Delta Ct}$ method [54], and relative expression was normalised to the abundance of sodium/potassium-transporting ATPase subunit alpha-like gene for mantle tissue and cAMP-dependent protein kinase catalytic subunit-like isoform X3 gene for gill tissue.

According with the $2^{-\Delta\Delta Ct}$ values, data for each tissue and condition selected were converted to natural logarithm and analysed using a student T-test (independent). Statistical comparison between the treatment and its respective control were performed, and statistical significances were set at $p < 0.05$. The $log_2FC$ was assessed using treatment vs control data to validate the expression of selected genes compared to the transcriptome results and by a linear regression. The software Statistica® 10.0 (StatSoft, Tulsa, OK, USA) was used to perform statistical analysis.

## Ethics statements

All the scallops used in this work were handled in accordance with the Official Mexican Standard protocols (NOM-062-ZOO-1999). *Argopecten ventricosus* juveniles were provided from the non-governmental association Noroeste Sustentable (NOS) and transported to aquarium tanks at CIBNOR with all the required permits from the federal agency CONA-PESCA. The animals used in this study were produced by Acuacultura Robles hatchery in captivity for experimental purposes. Organisms were kept in optimal culture conditions to avoid stressful conditions and no harmful effect has been detected using HDIC in marine organisms.

According to the Internal Committee for the Care and Use of Laboratory Animals (CIC-UAL) recommendations, the number of sampled organisms contemplated "the rule of maximizing information published and minimizing unnecessary studies". In this sense, for this work, we considered the minimum number of organisms needed to obtain a high-quality transcriptome.

## Results

### *De novo* assembly and functional annotation of the juvenile scallop *Argopecten ventricosus* transcriptome

To evaluate the effect of the immunomodulatory compounds on gene regulation, 20 libraries from juvenile Catarina scallops were considered and treated with the four HDIC (Fig 1B) versus control C1 (two libraries for experimental condition of each tissue) plus 12 libraries using NexSeq illumina in paired-end mode (2 X 150). A total of 423 million raw pair reads were generated. After the process of filtering reads, 372 million high-quality pair reads were used for the *de novo* transcriptome assembly. The final assembly generated 191 432 transcripts with a maximum length of 36 697 bp, minimum length of 184 bp, an average length of 863 bp and N50 value of 1123 bp (Fig 2A). The transcript size distribution showed a higher abundance of transcripts from 100 to 600 bp and lower abundance of transcripts from 2000 to 2200 bp (Fig 2B). The analysis of transcriptome completeness using the BUSCO tool against the database of Metazoa odb9, which consisted of benchmarking universal single-copy of conserved orthologous genes, showed that the percentage of complete and fragmented genes was 79.4% and 17%, respectively (Fig 2D).

Functional annotation results showed that 62 548 (32%) of the transcripts were annotated with at least one database. The *Mizuphecten yesseniosis* genome was the database that allowed the highest number of annotated transcripts (60157 hits). *M. galloprovincialis* genome and KEGG database showed the lowest number of annotated transcripts (15 425 and 17 487, respectively) (Fig 2A). The distribution of the BLAST top hits against the non-redundant

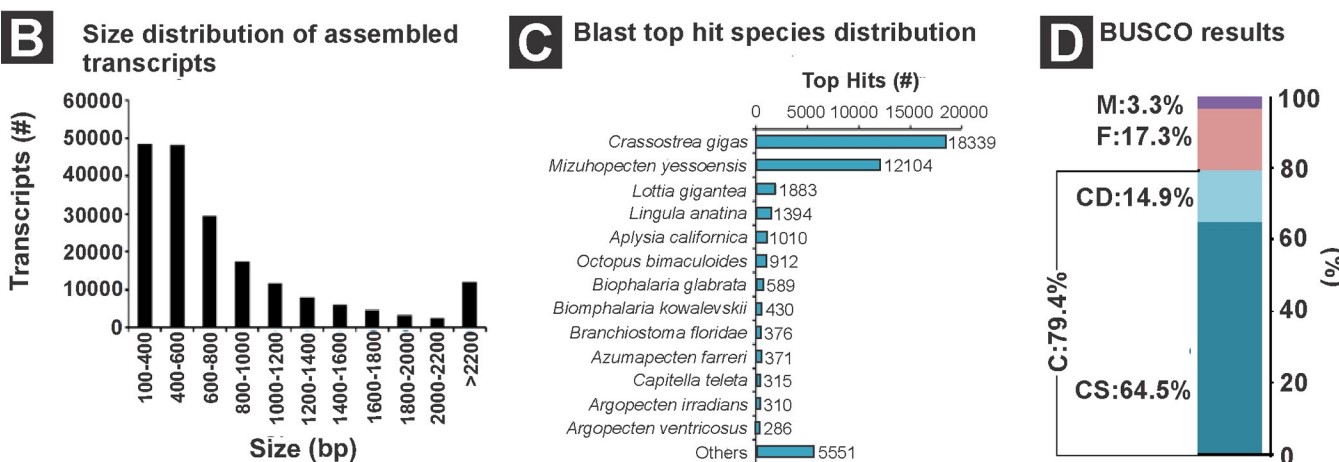

**A** Assembly statistics and functional annotation of the *A. ventricosus* transcriptome

| Sequencing results | | Assembly results | | Annotation results | | | |
|---|---|---|---|---|---|---|---|
| Raw reads pairs | 423,656,502 | Transcripts | 191,432 | NR | 43.541 | *Mytilus galloprovincialis* | 15,425 |
| Filtered read pairs | 372,879,514 | Average length of transcripts (bp) | 863 | GO mapped | 47,313 | *Crassostrea gigas* | 46,001 |
| Read legenth (bp) | 150 | N50 (bp) | 1,123 | KO mapped | 17,487 | *Crassostrea virginica* | 45,630 |
| | | Largest transcript (bp) | 36,697 | SwissProt | 33,265 | *Mizuhopecten yessoensis* | 60,157 |
| | | Smallest transcript (bp) | 184 | RefSeq | 46,404 | | |

**B** Size distribution of assembled transcripts

**C** Blast top hit species distribution

**D** BUSCO results

**Fig 2. Summary of the *novo* assembled transcriptome of *Argopecten ventricosus juveniles*.** (A) Assembly statistics and Functional Annotation of the scallop *A. ventricosus* transcriptome; (B) Size distribution of assembled transcripts; (C) Blast top hit species distribution; (D) BUSCO results. C: complete; CS: complete single-copy; CD: complete duplicate; F: fragmented, and M: missing.

protein (NCBI) database showed that the species with the highest number of top hits were the oyster *Crassostrea gigas* (18 339 hits) and the pectinid *Mizuphecten yesseniosis* (12 104 hits) (Fig 2C).

As a result, 47 313 transcripts were annotated from the functional annotation of the *A. ventricosus* juvenile transcriptome in Blast2go. A total of 29 116 transcripts were assigned to some biological processes, and the most represented were related to biosynthesis (2749), signal transduction (2 711) and cellular nitrogen compound metabolic process (2485) (Fig 3A). For molecular functions 30 274 transcripts were annotated. The most represented molecular function categories were related to ion binding (6662), oxidoreductase activity (2233) and transmembrane transporter activity (2076) (Fig 3A). In the category of cellular components 19 886 transcripts were recorded. The cellular component with higher number of transcripts corresponded to protein containing complex (3305), cytoplasm (1329) and intracellular (1246) components (Fig 3A). The KEGG annotation mapped 17 487 transcripts and showed the metabolic pathways mostly related to biosynthesis of secondary metabolites (222), thermogenesis (139) and endocytosis (128) (Fig 3B).

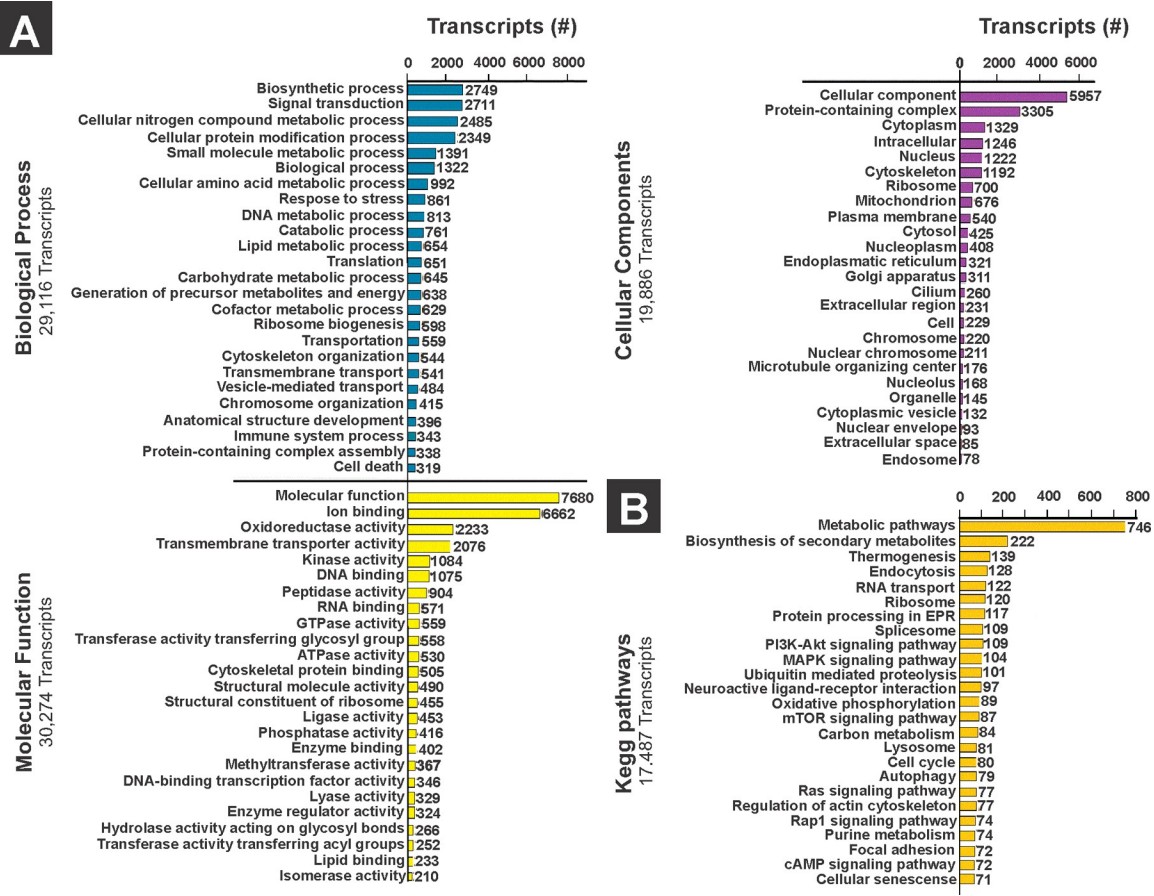

**Fig 3. Top 25 gene ontology terms and metabolic pathways from *Argopecten ventricosus* juvenile annotation.** (A) Histogram of the gene ontology classification showing the top 25 biological process (blue bars), molecular function (yellow bars), and cellular component (purple bars); (B) Histogram of the top 25 metabolic pathways from the annotation of the *A. ventricosus* juvenile transcriptome using KEGG database (orange bars). Graphics show the number of transcripts that participate in each classification.

## Transcriptional response to highly-diluted immunomodulatory compounds (HDIC)

Mantle was the tissue with the highest DEG number, particularly in organisms from T1 (3933; 2755 up- and 1178 down-regulated). The mantle of scallops treated with T2 (2002; 1021 up- and 981 down-regulated) recorded the lowest DEG number (S2 Table). In gill, the tissue with less DEG, T3 (1744; 664 up- and 1080 down-regulated) recorded the highest DEG number and T1 (1125; 365 up- and 760 down-regulated) the lowest (S2 Table).

The Venn diagram showed that juvenile scallops had a specific response to each treatment as most of the DEG were not shared between treatments. In most of the cases, the mantle recorded a higher DEG number by the treatment effect compared to gill. The scallops that recorded the highest number of up- and down-regulated DEG in mantle were from T1/C1 (1841 up, 491 down) while the lowest number of up-regulated DEG were recorded in scallops from T2/C1 (257 up, 275 down) (Fig 4A). The DEG profiles in mantle were clustered, showing that T3 and T4 were clustered in the same group and T1 and T2 in other groups separately (Fig 4B).

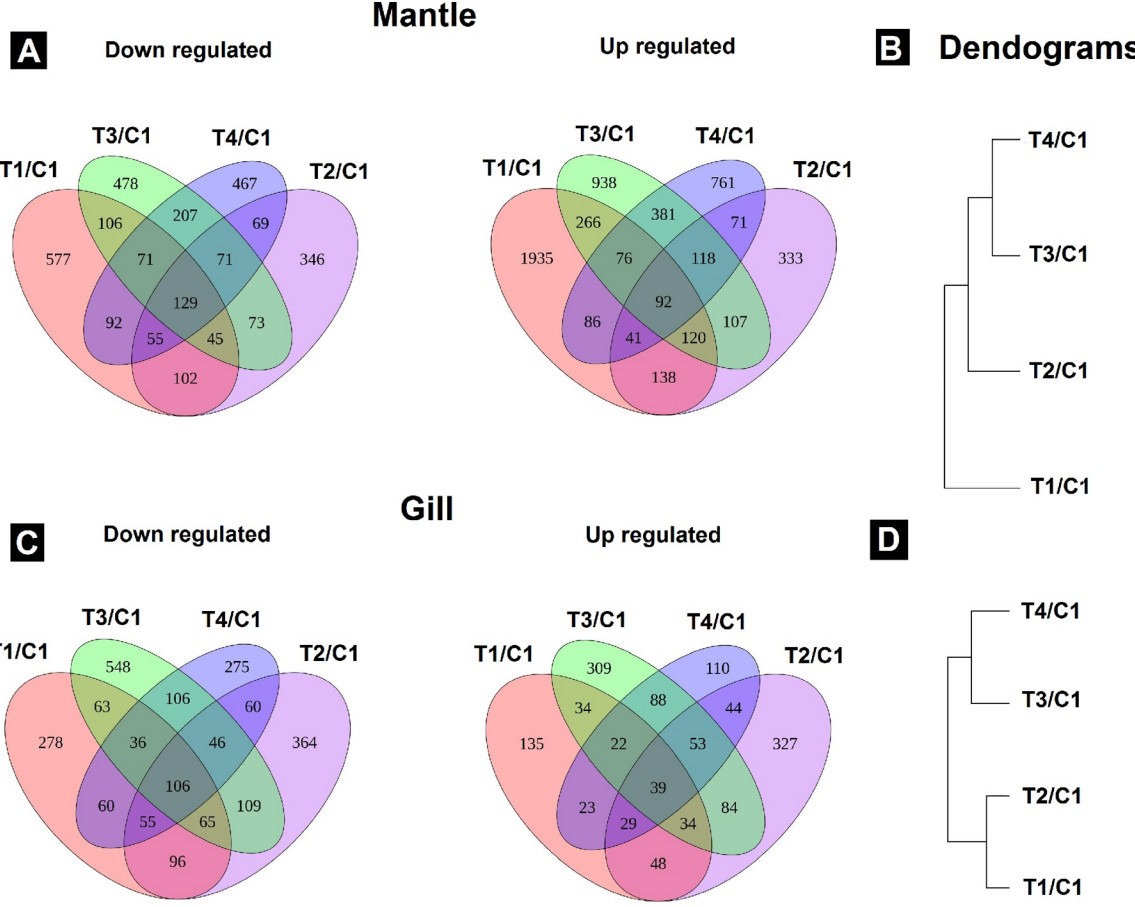

**Fig 4. Venn diagrams of differentially expressed genes (DEG).** Up- and down-regulated DEG in mantle (A) and gill (C) tissue of *Argopecten ventricosus* juveniles treated with highly-diluted immunomodulatory compounds (HDIC). The dendogram shows the relationship between gene expression by hierarchical clustering by treatment in mantle (B) and gill tissue (D). T1: *Vibrio parahaemolyticus* and *Vibrio alginolyticus* lysate 1D; T2: *V. parahaemolyticus* and *V. alginolyticus* lysate 7C; T3: Silicea terra® and Phosphoric acid® (Similia®, CDMX,MX) 7C; T4: Vidatox® (Labiofam®, Habana, Cuba) 31C, and control treatment C1: hydro-alcoholic solution (1%).

In gill, treatments with the highest number of up- and down-regulated DEG were recorded in scallops from T2/C1 (248 up, 309 down) and T3/C1 (235 up, 483 down). The lowest number of up- and down-regulated DEG in gills were recorded in T4/C1 (94 up, 245 down) (Fig 4C). T1 and T2, formulated by *V. parahaemolyticus* and *V. alginolyticus* lysate in different concentration, were clustered in the same group. T3 (Silicea terra® and Phosphoric acid®) and T4 (Vidatox®, formulated by commercial brands, were clustered in a different group (Fig 4D).

## Co-regulation of gene expression and functional enrichment analysis

Hierarchical clustering analysis of the differentially expressed transcripts allowed identifying groups of co-expressed transcripts for each tissue and treatment. Furthermore, the GO biological processes and metabolic pathways enriched for each group were analysed (Figs 5 and 6). In mantle, the clustering analyses showed 20 clusters; from these clusters seven were enriched by GO terms (*p*-value < 0.01). Interestingly, three groups were related to oxidative phosphorylation (2,12,14), and four groups (cluster 13, 15, 16 and 20) were enriched in biological processes related to non-self-recognition and immune response processes. Cluster 13 comprised five transcripts related to focal adhesion and five related to endocytosis (Fig 5). Most of the DEG from cluster 13 in mantle were down-regulated mainly by the effect of T4/C1 (mean $Log_2FC$ -3.7) (S1 Fig). A total of five transcripts from cluster 15 were related to apoptosis process. DEG from cluster 15 were mainly up-regulated in all treatments with a higher number of up-regulated DEG in T3/C1 (mean $Log_2FC$ 5.8). In cluster 16, 44 transcripts were related to pathogenic *Escherichia coli* infection pathway, 31 to cytoskeleton organization and eight to immune response. All treatments from cluster 16 in mantle were mainly up-regulated with the highest number of up-regulated DEG in T1/C1 (mean $Log_2FC$ 2.5) (S1 Fig). Cluster 20 comprised 65 transcripts related to signal transduction and 18 to pathogenic *E. coli* infection pathway. Cluster 20 showed up- and down- regulated DEG with the highest number of up-regulated DEG in T4/C1 (mean $Log_2FC$ 0.7) and the highest number of down-regulated DEG in T1/C1 (mean $Log_2FC$ -0.5) (Figs 5 and S1). In mantle the enrichment of clustered co-expressed genes were in accordance to the enrichment of all DEG as shown in (S2 and S3 Figs).

On the other hand, 17 groups were generated in gill, of which six showed enriched GO terms (*p*-value < 0.01, Fig 6). The enrichment analysis showed that three groups (6, 7 and 12) were related to oxidative phosphorylation and three (clusters 5, 8 and 16) were related to non-self-recognition and immune response (Fig 6). In cluster 5 from gill tissue, 11 transcripts were related to cellular response to stimulus (Fig 6). Scallops from cluster 5 showed a DEG down-regulation mainly by effect of T1/C1 (mean $Log_2FC$ -7) (S1 Fig). Cluster 8 comprised 10 transcripts related to NF-kappa B signalling pathway, 44 to signal transduction and 17 to stress response. In cluster 8 the DEG were mainly down-regulated in all treatments with a higher number of down-regulated DEG in T2/C1 (mean $Log_2FC$ -1). A total of 13 transcripts from cluster 16 were related to transmembrane transport in gill tissue. Transcripts for all the treatments in gill were mainly up-regulated in cluster 16 with the highest number of up-regulated DEG in T1/C1 (mean $Log_2FC$ 2.5) (Figs 6 and S1). As recorded in mantle, the enrichment of clustered co-expressed genes from gill tissue were in accordance to the enrichment of all DEG shown in (S2 and S3 Figs).

## Identification of genes associated to immune response regulated by effect of highly-diluted immunomodulatory compounds (HDIC)

Since HDIC have shown to improve the ability of these organisms to cope with pathogen infections [20], probably through immune system modulation [12,21], the transcripts related to the immune system were identified and their expression levels analysed in response to the

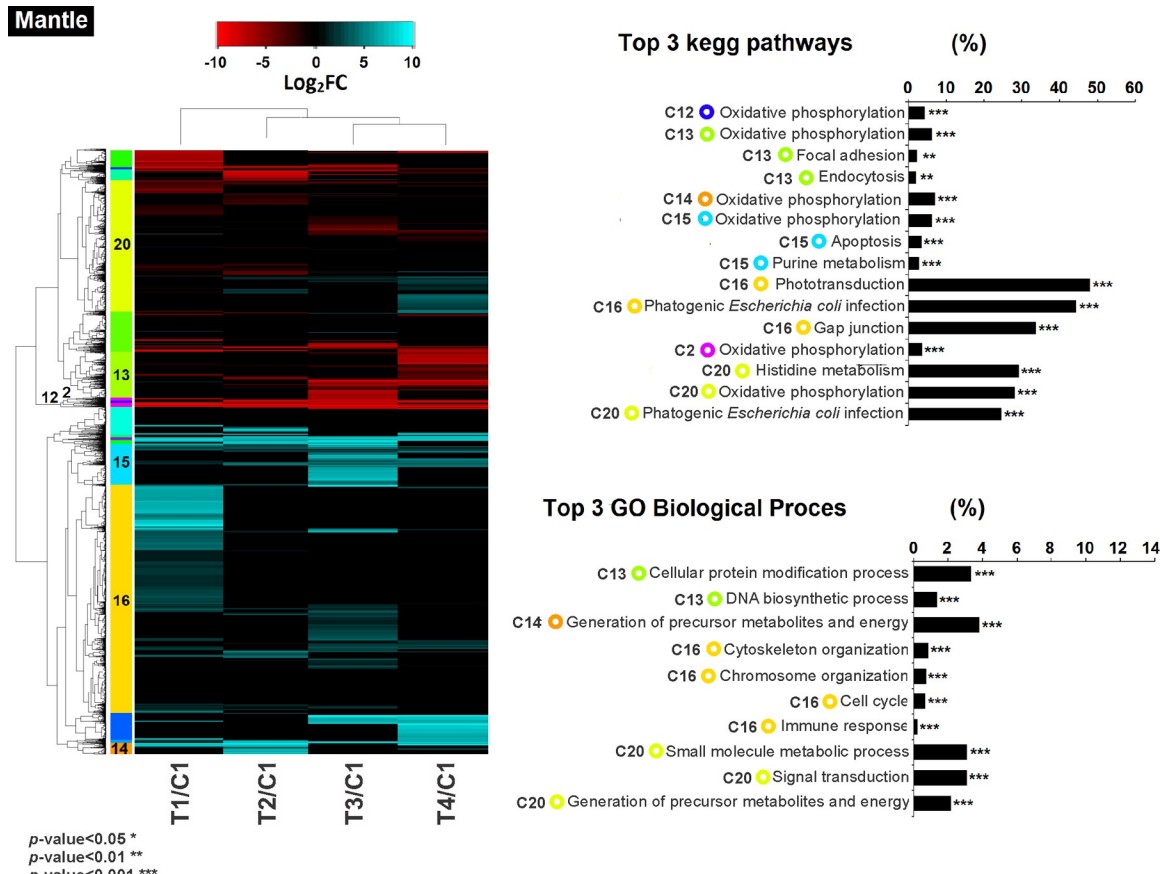

**Fig 5. Heatmap showing clusters of co-regulated genes with their expression profiles in mantle tissue of the Catarina scallop *Argopecten ventricosus* juveniles treated with highly-diluted immunomodulatory compounds (HDIC).** The dendrogram shows the relationship between gene expression by hierarchical clustering. Clusters in differentially expressed coloured boxes at the left indicate gene clusters with similar expression profiles. The colour key indicates the $Log_2FC$ of the DEG (FDR < 0.01) ranging from cyan blue for most up-regulated to red for most down-regulated genes. Cluster with functional enrichment with GO or KEGG database with more than five transcripts participating in the process are shown in the right side graph (C2:Cluster 2, C9:Cluster 9, C10:Cluster 10, C12: Cluster 12, C13:Cluster 13, C14:Cluster 14, C15:Cluster 15, C16:Cluster 16, C20:Cluster 20). Circles show the colour of their corresponding co-regulated gene cluster. Contrast conditions were clustered to identify similarity pattern in the DEG between treatments. T1: *V. parahaemolyticus* and *V. alginolyticus* lysate 1D; T2: *V. parahaemolyticus* and *V. alginolyticus* lysate 7C; T3: Silicea terra® and Phosphoric acid® (Similia®, CDMX, MX) 7C; T4: Vidatox® (Labiofam®, Habana, Cuba) 31C, and control treatment C1: hydro-alcoholic solution (1%).

treatments. Based on the functional annotation and GO and KEGG categorisation, a total of 193 transcripts were identified related to the non-self-recognition receptor, internalisation and immune system. The expression analyses showed that the tissue with higher DEG number related to immune response was mantle (175 transcripts) compared to gill (28 transcripts) (S3 and S4 Tables). The representative metabolic pathways with higher DEG numbers were phagosome, endocytosis and NF-kappa B signal, but non-self-recognition receptors were also found.

In mantle, higher DEG related to immune response were recorded in scallops from T1 (C1 (up: 58, down: 11) and T3/C1 (up: 56, down: 13), and lower in T4/C1 (up: 17, down: 16). The functional analyses of these transcripts showed that the metabolic pathways with a higher number of up-regulated DEG in T1/C1 were NF-kappa B signal pathway (10), phagosome (9) and endocytosis (10) in mantle. Also, T1/C1 up-regulated lysosome pathway (4), toll like receptors (1) and heat shock proteins (2) but with a lower DEG number. Mainly down-regulated metabolic pathways in mantle recorded in organisms from T1/C1 were related to the

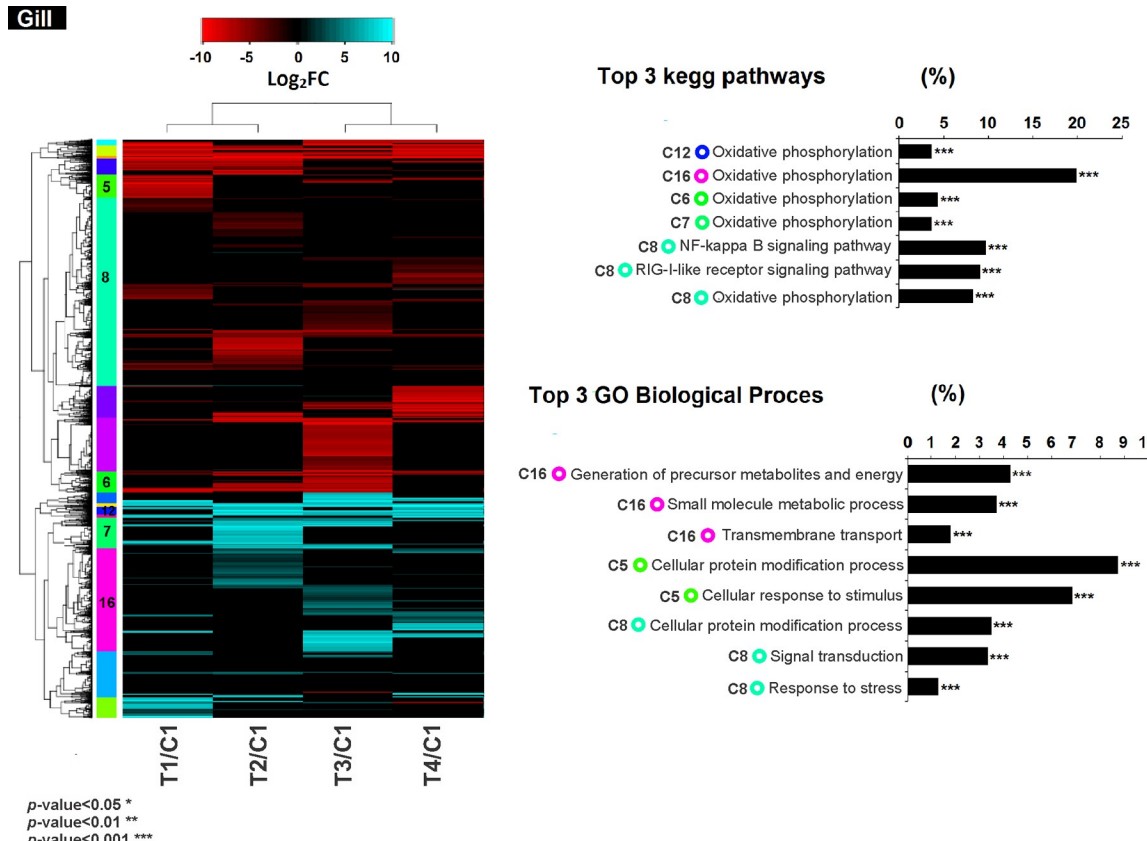

**Fig 6. Heatmap showing clusters of co-regulated genes with their expression profiles in gill tissue of the Catarina scallop** *Argopecten ventricosus* **juveniles treated with highly-diluted immunomodulatory compounds (HDIC).** The dendrogram shows the relationship between gene expressions by hierarchical clustering. Clusters differentially expressed in colour boxes at the left indicate gene clusters with similar expression profiles. The colour key indicates the Log$_2$FC of the DEG (FDR < 0.01) ranging from cyan blue for most up-regulated to red for most down-regulated genes. Cluster with functional enrichment with GO or KEGG database with more than five transcripts participating in the process are shown in graphs on the right side (C5: Cluster 5, C6: Cluster 6, C7: Cluster 7, C8: Cluster 8, C10: Cluster 10, C12: Cluster 12, C14: Cluster 14, C15: Cluster 15, C16: Cluster 16). Circles show the colour of their corresponding co-regulated gene cluster. Contrast conditions were clustered to identify the similarity pattern in the DEG between treatments. T1: *Vibrio parahaemolyticus* and *Vibrio alginolyticus* lysate 1D; T2: *V. parahaemolyticus* and *V. alginolyticus* lysate 7C; T3: Silicea terra® and Phosphoric acid® (Similia®, CDMX,MX) 7C; T4: Vidatox® (Labiofam®, Habana, Cuba) 31C, and control treatment C1: hydro-alcoholic solution (1%).

phagosome (6), which also had up-regulated genes. T3/C1 allowed up-regulation of DEG related to endocytosis (up: 8, down: 3), focal adhesion (7) and phagosome (up: 7, down: 3) in organism's mantle. The highest numbers of down-regulated DEG in T3/C1 were observed in MAPK signalling pathway (4). In scallops from T4/C1, the metabolic pathways with higher numbers of up-regulated DEG were observed in hematopoietic cell linage (2) and melanogenesis (3) pathway; the highest numbers of down-regulated DEG were recorded in focal adhesion (3) and endocytosis (up: 2 down: 5) pathway (Fig 7 and S3 Table).

In gill, most of the pathways were down-regulated; the treatment with the highest DEG was T2/C1 (up: 1, down: 15), and the one with the lowest DEG was T4/C1 (up: 1, down: 3). As in mantle, scallops from T2/C1 showed a higher number of down-regulated DEG (15) in gill compared with the up-regulated DEG (1). The metabolic pathways with higher numbers of down-regulated DEG in T2/C1 were recorded in the NF-kappa B signal pathway (4) and endocytosis pathway (up: 1, down: 2). The gill of the organisms treated with T2/C1 up-regulated only one

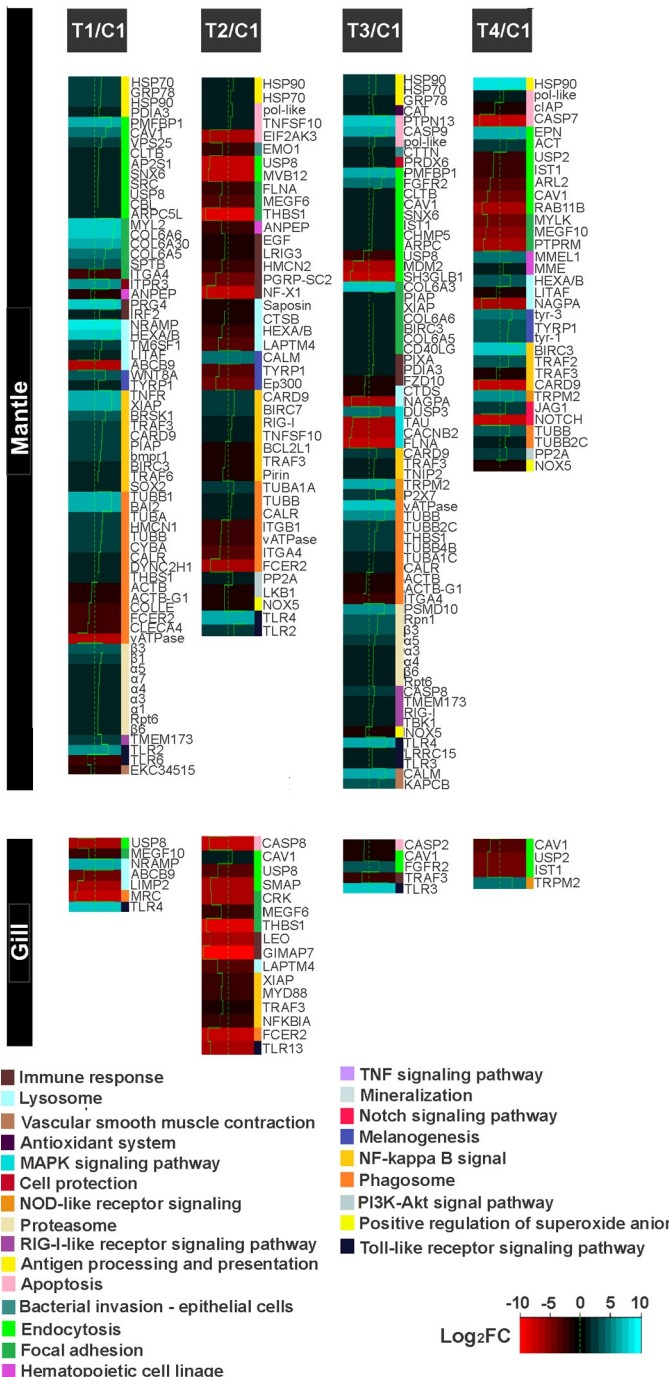

**Fig 7. Differentially expressed genes (DEG) modulated for each treatment and tissue (mantle and gill) related to immune response, signal internalisation and non-self-recognition in *Argopecten ventricosus* juveniles treated with highly-diluted immunomodulatory compounds (HDIC).** DEG are grouped in to metabolic pathways categories. T1: *Vibrio parahaemolyticus* and *Vibrio alginolyticus* lysate 1D; T2: *V. parahaemolyticus* and *V. alginolyticus* lysate 7C; T3: Silicea terra® and Phosphoric acid® (Similia®, CDMX,MX) 7C; T4: Vidatox® (Labiofam®, Habana, Cuba) 31C, and control treatment C1: hydro-alcoholic solution (1%).

gene, and it was related to endocytosis. The metabolic pathways with higher numbers of up-regulated DEG in T4/C1 were observed in NOD-like receptor signalling (1) (Fig 7 and S4 Table).

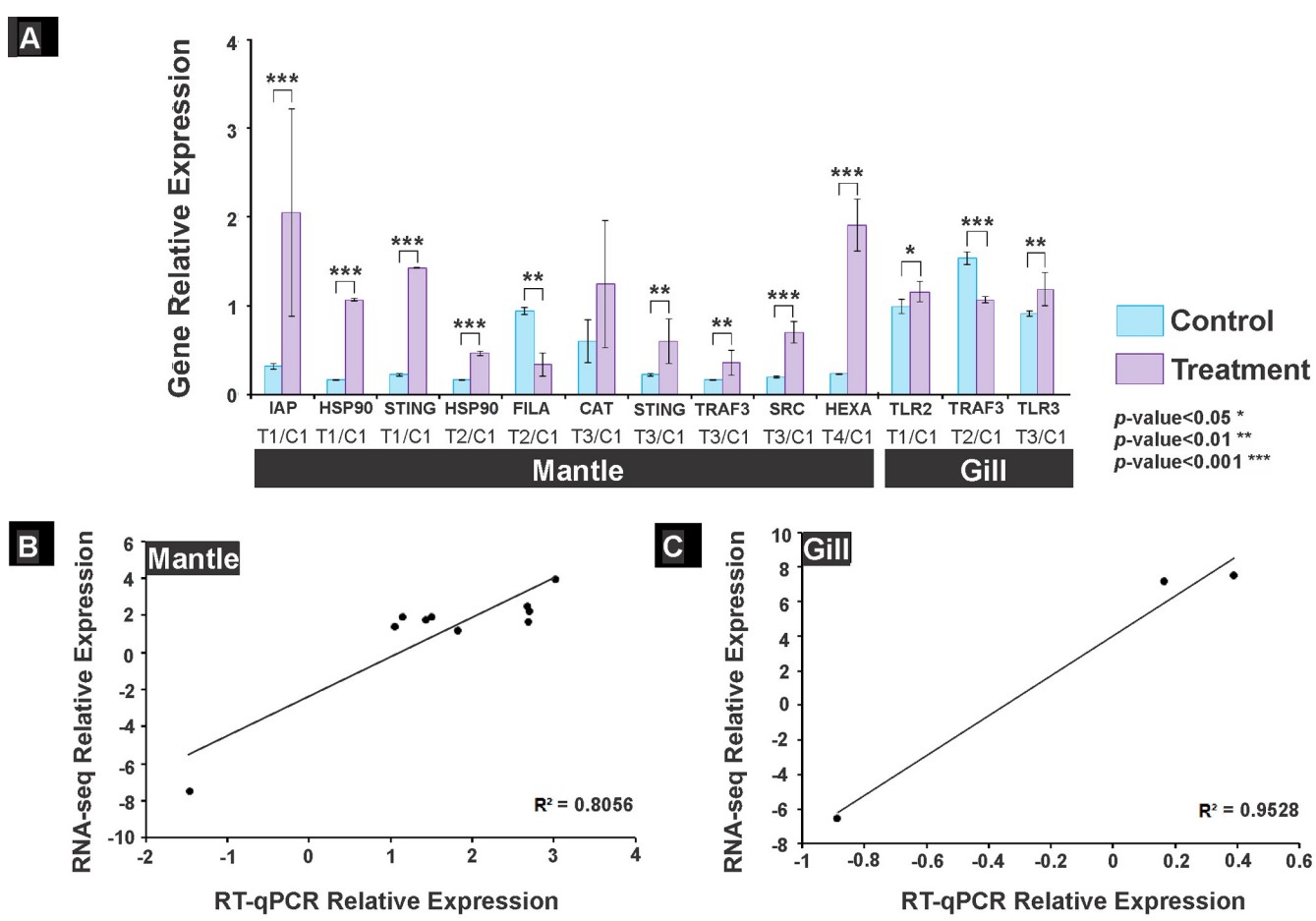

**Fig 8. Real time qPCR validation of *Argopecten ventricosus* juvenile transcriptome data for selected genes.** (A) Relative expression normalised to the abundance of sodium/potassium-transporting ATPase subunit alpha-like gene in mantle tissue and correlation between Log₂FC transcriptome data and RT-qPCR data in mantle; (B) Relative expression normalised to the abundance of cAMP-dependent protein kinase catalytic subunit-like isoform X3 gene in gill tissue and correlation between Log₂FC transcriptome data and qPCR data in gill. IAP: Baculoviral IAP repeat-containing protein 3-like, HSP90: Heat shock protein 90, ERIS: Endoplasmic reticulum interferon stimulator, SRC: Src substrate cortactin-like isoform X1, HEXA: Beta-hexosaminidase, CAT: Catalase, TRAF3: TNF receptor-associated factor 3, FILA: Filamin-A-like isoform X3, TLR2: Toll-like receptor 2, TLR3: Toll-like receptor 3. T1: *Vibrio parahaemolyticus* and *Vibrio alginolyticus* lysate 1D; T2: *V. parahaemolyticus* and *V. alginolyticus* lysate 7C; T3: Silicea terra® and Phosphoric acid® (Similia®, CDMS, MX) 7C; T4: Vidatox® (Labiofam®, Habana, Cuba) 31C, and control treatment C1: hydro-alcoholic solution (1%).

## Validation of RNA-seq data by qPCR analysis

Ten relevant DEG related to the immune response and regulated by the four treatments in this study were selected to validate the transcriptome data by RT-qPCR analysis. All contrasts were made using the control group as a reference; relative expression was normalised to the abundance of sodium/potassium-transporting ATPase subunit alpha-like gene in mantle tissue and cAMP-dependent protein kinase catalytic subunit-like isoform X3 gene in gill tissue because they were the most stable genes among the experimental conditions. The up- and down-regulated genes selected from the transcriptome had the same pattern in RT-qPCR analysis. IAP, HSP90; ERIS genes significantly ($p < 0.001$) up-regulated in organism mantle by effect of T1. In T2 gene HSP90 ($p < 0.001$) from the mantle significantly up-regulated and FILA ($p < 0.01$) down-regulated. The genes ERIS, TRAF3 ($p < 0.01$) and SRC ($p < 0.001$) up-regulated in the organism mantle from T3, and in T4 they promoted up-regulation of HEXA gene ($p < 0.001$) (Fig 8A). In gill T1 up-regulated ($p < 0.05$) the expression of TLR2 gene; T2 promoted the up-

regulation ($p < 0.001$) of TRAF3 gene; T3 regulated the expression of TLR3 gene ($p < 0.01$) (Fig 8A). Finally, the correlation between the expression of the genes in mantle and gill using Log$_2$FC data from RT-qPCR and RNA-seq analyses was 0.80 and 0.94, respectively (Fig 8B and 8C), which were considered good.

## Discussion

Most of the transcriptomes assembled in marine bivalves did not report fragmentation percentage in their results; however, fragmentation is known to be typical in *de novo* transcriptome assemblies [55]. The assembly statistics in this study was according to most published marine bivalve transcriptomes referred by the authors as high quality assembly [56, 57,58] and showed a higher quality assembly compared to others [27,59,60].

The effectiveness of the use of HDIC in bivalves to improve their survival against pathogen infection challenges has been widely demonstrated, our work team has proven that the application of HDIC formulated with scorpion venom, Phosphoric acid®, Silicea terra®, sodium metasilicate and *Vibrio* lysate (*V. alginoliticus-V. parahemolityics*) strengthened marine organisms (bivalves, fish and shrimp) self-defense, allowing them to survive infection challenges, reduce parasite proliferation, increase energy reserves and growth; even some of them outperformed antibiotics when they were used prophylactically in *A. ventricosus* culture [20,21,61,62].

The efficiency of these HDIC to protect marine organisms against infections has been attributed to increases of the antioxidant system response, haemocyte count, energetic reserves and enzymes related to food assimilation [20,21,63]. Furthermore, attenuated *V. splendidus* cells activated the immune response in the scallop *Argopecten purpuratus* [18] and micro-sized silica [64], phosphorus [65] and toxins [66] activating antioxidant response, which allowed organisms to face pathogen infections by promoting free radical eradication. This information considered that the evaluated HDICs may have the potential to modulate immune response at high and low concentrations; however, the mechanisms regulated at low dilutions have not been elucidated yet. Information about how genes and mechanisms are being regulated in marine organisms treated with HDIC is scarce. In this sense, interestingly, the results from the transcriptomic analysis in this study suggested that the immune response was activated by most of the HDIC but not like conventional immunostimulants, such as polysaccharides, nutrients, oligosaccharides, herbs, antibacterial peptides and microorganisms [11] used at higher concentrations, which increased production of bactericidal and cytotoxicity activities, lysozyme and antimicrobial peptides [11,67,68]. In this study, HDIC regulated transcripts associated to recognition of non-self-molecules and internalisation of particles suggested that HDIC were not able to trigger all the immune response mechanisms due their high dilution. This result may explain those [61] in organisms of *S. rivoliana* treated with HDIC where only the genes IL-1β and MyD88 up-regulate before the challenge with *V. parahaemolyticus*. The results in this study suggested that the non-self-recognition system activation by effect of HDIC improved the response against infections. When the defense mechanisms were activated before the infection process, the organisms had more probabilities to successfully overcome a disease, which was the principle of the immunostimulants used in aquaculture [11].

It is worth to highlight that most of the DEG related to non-self-recognition, internalisation and immune response were detected in mantle compared to gill tissue. This result can be explained because mantle and mucosal epithelial cells are one of the first barriers that are in contact with the environment, and epithelial cells have been recognised as the first line of defense against organic, inorganic and pathogen intruders [69]. Epithelial cells are able to endocytose biotic and abiotic particles and activate signals to proliferate haemocytes, the

principal effector of the immune response [70], which in turn can also eliminate intruders by endocytosis and enhance immune response [70]. Both epithelial cells and haemocytes can detect intruders by many pattern recognition receptors (PRR) that allow organisms to detect potential danger signals [69]. Thus, the up-regulated transcripts related to non-self-recognition, internalisation and immune response in mantle may be due to the participation of epithelial mantle cell, migrating haemocytes or both. Future research should clarify and corroborate which cells are interacting and recognising the HDIC.

The recognition mechanisms of marine bivalves that detect non-self-molecules are integrated by the PRR, which are activated by pathogen-associated molecular patterns (PAMPs), endogenous ligands (e.g. HSP70) and damage associated molecular patterns (DAMPs). The activation of the PRR allow organisms to activate the immune response depending on the stimuli received [59]. In agreement with the previous information, the results in this study showed that the up-regulated transcripts are related to PRR from the toll-like receptors family, which are highly conserved in the animal kingdom and have proven to activate the immune response in marine bivalves [59]. Specifically, this study detected the induction of TLR2 by the effect of T1 and T2, and TLR4 in T2 and T3. Moreover, T3 up-regulated TLR3. The activation of TLR2 and TLR4 in T1 and T2 was attributed to their formulation (*V. alginoliticus* and *V. parahemolityics*) because TLR2 and TLR4 are Pattern recognition receptors (PRR) located in plasma membrane and mainly recognise bacterial PAMPs, such as lipopolysaccharides [71] that can be found in gram negative bacteria like *V. alginoliticus* and *V. parahemolityics* [72]. In bibliography no data was found between the interaction of TLR3 and Silicea terra® or Phosphoric acid®, which were the components in T3, but we only know that TLR3 is a cytosolic receptor that detects non-self-nucleic acids [73]. In this sense, this study has reported for the first time that TLR3 is associated to the exposure of highly-diluted Silicea terra® or Phosphoric acid®. Additionally, these results showed that T3 formulated by Silicea terra® and Phosphoric acid® promoted the induction of the TLR4 gene. Accordingly, evidence that $SiO_2$ nanoparticles are able to increase the TLR4a gene expression has been observed in Dino zebrafish embryos [74], and *S. rivoliana* juveniles treated with Silicea terra® - Phosphoric acid® up-regulated MyD88 transcript only when organism were challenged. It should be noted that MyD88 regulation is highly related to TLR4 [75], which suggested that the up-regulation of TLR4 in T3 was associated to Silicea terra® action. In T4, formulated with Vidatox®, no PRR regulated were observed, which may explain the low regulation of transcripts related to immune response. The information above suggested that HDIC had been detected mainly through toll-like receptors in treatments T1, T2 and T3, while Vidatox® may have detected them by an alternative route in T4.

When small intruders (e.g. Bacteria, LPS) are detected via PRR, many mechanisms can be activated to respond against the potential danger signals, such as PAMPs and DAMPs [70]. The mechanisms activated in this study were dependent on each treatment and their dilutions. Treatments T1 and T2 mainly modulated the expression of transcripts related to endocytosis, lysosome, phagosome, proteasome and NF-kappa B pathways. Interestingly, T1, which had a higher concentration of bacterial compounds than T2, up-regulated transcripts associated to those processes while T2 regulated them negatively. Other authors have reported that the presence of PAMPs activated endocytosis, lysosome, and phagosome mechanisms [69,70] to destroy potentially harmful molecules or disease-causing microorganisms. The activation of those mechanisms can also activate the signalling cascade to express immune related genes [69,70] via NF-kappa B pathway signal [76]. This result suggested that the organisms treated with T1 recognised and responded to the presence of PAMPs or DAMPs in the HDIC formulated by *V. parahaemolyticus* and *V. alginolyticus* lysate.

In this study, most of the transcripts that up-regulated in NF-kappa B pathway by T1 were modulators at the first level of the immune response and effectors of the anti-apoptosis process. In this sense, T1 allowed activating a conservative response. The results showed that T1 also up-regulated genes related to sense microbial viability (STING), which allowed organisms to detect living bacteria [77], bind Vibrio protein (CALR) that can be released from the cell and neutralise vibrio [78,79] and heat shock proteins (HSP70 and HSP90), related to showing antigen (adaptive immune system) [80], protein fold [81], activating Toll-like receptors [82] and mediating immune response in bacterial challenges [83]. The above suggested that T1 allowed the activation of pathways related to recognition, neutralisation, internalisation and destruction of *Vibrio*. Although T2 did not activate transcripts associated to mechanisms of endocytosis, lysosome, phagosome, proteasome and NF-kappa B pathways, it has been successfully used in strengthening marine organisms against pathogens [12,13,61]. These results showed that the induction of TLRs in scallops by T2 may have been enough to protect organisms against pathogens. Because T2 was highly diluted compared to T1, the concentration of molecules from the *V. parahaemolyticus* and *V. alginolyticus* lysate might have not been enough to activate the internalisation and destruction of non-self-molecule mechanisms; this situation allowed organisms to tolerate the presence of these molecules and increase the expression of the transcripts related to detection mechanisms of potential danger signals, such as TLRs, which may imply a lower energetic cost for the organisms. Perhaps, as previously reported, PAMPs used at high-doses induced tolerance while at lower ones they allowed trained sensitisation, and in very low doses, they did not activate immune response [84]. Higher dilutions from the HDIC used in this study were in accordance with literature [84], which proposed that very low doses of PAMPs did not activate innate immune response. However, for the first time in *Argopecten ventricosus* this study detected the up-regulation of transcripts related to TLRs without activating the immune response. Contrary to very low doses, the low doses of HDIC in this study showed an activation of detection, neutralisation and internalisation responses against intruders without activating all the immune response mechanisms. The above suggests that a prolonged exposure time to low PAMPs doses may also generate some tolerance because the activation of immune response implies high ATP amounts. The reason why innate cells depend on PAMPs dose and exposure time may be explained by the energetic cost; if organisms sense a very low or low concentrations of PAMPs, the energetic balance is maintained; the organisms can invest energy in processes related to self-protection depending on the dose and time of exposure to the stressor (e.g. LPS).

Interestingly, organisms treated with T3 (Silicea terra® - Phosphoric acid®) induced transcripts related to bacterial challenge (STING, CALR), antioxidant system (CAT, PRDX6) [59,85], cell migration (CTTN) [86], amplification of immune response (CD40LG, found in adaptive response) [87], and the receptor associated to actin-assembly machinery on the cytoplasmic side of the phagosome (P2X7) [88]. In previous publications, Silicea terra® and Phosphoric acid® had recorded an improvement of *A. ventricosus* juvenile growth, antioxidant activity [21] and survival when organisms were challenged against pathogenic bacteria [20]. Moreover, silicic acid, which has been reported in silica solutions [89], may activate catalase (CAT) and superoxide dismutase (SOD) activities in mouse brain from organisms intoxicated with aluminium [90]; in *S. rivoliana*, the use of Silicea terra® and Phosphoric acid® increased survival when organisms were challenged against pathogenic bacteria [61]. These results suggested that the activation of non-self-recognition, endocytosis and antioxidant system by T3 was the reason for survival when organisms were challenged.

As mentioned before, T4 behaved differently compared to the rest of the treatments and mainly activated the melanogenesis and haematopoietic processes and a gene associated to calcium transport (TRPM2), which have been linked to immune response when organisms are

challenged against bacteria [70,71]; however, the immunomodulatory effect of Vidatox® was not clear. Vidatox® has demonstrated to increase SOD activity and survival of *L. vannamei* when it has been challenged against pathogenic bacteria [91], but it has also been related to proliferation of hepatocellular carcinoma in cultured mouse cells [92]. As in the other treatments, oxidative phosphorylation in mitochondria was modulated by the effect of T4. The activation of the immune response may be modulated by alternative routes as inflammasomes that are activated by DAMPs [71] and highly related to the production of mitochondria ROS (mROS), which have been implied in the activation of the immune response [71,88]. This result could explain the hepatocellular carcinoma proliferation in cultured mouse cell when Vidatox® was used since proliferation of mROS may cause oxidation damage in cancer cells [93]; the positive effect recorded in challenged shrimp, as mROS, may also have been related to the activation of immune response and killing pathogens [71]. Thus, T4 may be more related to the activation of immune response by mitochondria signalling. Interestingly, T4 was formulated by the most diluted compound, suggesting that molecules highly diluted mainly act by their magnetic charge [14], sensed by traditional molecular receptors or photoelectrochemical sensing system activated by ultra-weak photo emission signals [15]. In *Arabidopsis thaliana* their defense mechanisms may be activated by the sound of caterpillars (*Pieris rapae*) eating their leaves [94] *via* mitochondria signalling; their mROS production has been implied in vibration recognition [95]. Since mitochondria have been related to the activation of the immune response by vibration recognition mechanisms, future research should analyse the effect of Vidatox®, taking into account the importance of mitochondria and oxidative phosphorylation process.

It is worth to notice that all the treatments modulated the oxidative phosphorylation process in mitochondria, an organelle that is not only related to vibration recognition but also plays a crucial role in supplying ATP since it is necessary for cell internalisation process, acidification or the phagosome, production of mROS for eradication of bacteria and activation of immune signalling cascades [71,88]. Nonetheless, the up- or down-regulated process could not be detected due to transcriptome fragmentation (17%), which diminished the robustness of expression estimates in fragmented genes [55].

This study used different HDIC, most of them showing up-regulation in the PAMPs recognition system, which may explain why marine organisms increased survival challenges against pathogens in previous studies [12].

## Conclusion

This study opens new insight into the activation of *Argopecten ventricosus* self-protection mechanism using HDIC, which allowed answering questions and opening the possibility of generating others about interactions between organisms and their exposure to low toxin concentration, contaminants, minerals, pathogens and other stress factors during a prolonged time. The results of this study implied that lower dilutions activated mechanisms of immune response in a higher level in a shorter period while higher dilutions activated only the first defense mechanisms in *A. ventricosus*, which may be dependent on the exposure time between the organisms and the HDIC. These results showed the complex dynamics between the non-self-recognition mechanisms of *A. ventricosus* juveniles and HDIC in a long time exposure. Additionally, mitochondria, besides its role in energy production, may play an important role in defense response activation and signalling process when organisms are exposed to HDIC. Furthermore, this is the first assembled transcriptome of the scallop *A. ventricosus* juveniles using RNA-seq technology that allows us to generate biological markers for future investigation of this important resource, whose populations are declining in Baja California Sur, México.

## Supporting information

**S1 Fig. Expression patterns for cluster with functional enrichment with Gene Ontology (GO) or Kyoto Encyclopaedia of Genes and Genomes (KEGG) data base.** Biological process or metabolic pathways with more than five transcripts are shown in graph with grey lines with the average datum behaviour represented by a red line.
(TIF)

**S2 Fig. Biological process enrichment of Gene Ontology (GO) categories based on the differentially expressed genes (DEG) in each analysed contrast in mantle and gill tissue.** The colour key indicates from beige to brown the increasing percentages of genes representing each up-regulated category and from beige to orange the increasing percentages of genes representing each down-regulated category. Blast2go specific filter was applied to the enrichment analysis to eliminate general categories. Contrasting conditions referred to all treatments evaluated vs control.
(TIF)

**S3 Fig. Top 25 metabolic pathways enriched (Kyoto Encyclopaedia of Genes and Genomes) categories based on the differentially expressed genes (DEG) in each analysed contrast in mantle and gill tissue.** The colour key indicates from beige to brown the increasing percentages of genes representing each up-regulated category and from beige to orange the increasing percentages of genes representing each down-regulated category. Blast2go specific filter was applied to the enrichment analysis to eliminate general categories. Contrasting conditions referred to all treatments evaluated vs control.
(TIF)

**S1 Table. Primers used to validate the transcriptome differentially expressed genes (DEG) analysis.**
(XLSX)

**S2 Table. Total differentially expressed genes (DEG) in mantle and gill from scallops treated with highly-diluted immunomodulatory compounds (HDIC).**
(XLSX)

**S3 Table. Differentially expressed genes (DEG) in mantle selected from enriched biological process and metabolic pathways related to non-self-recognition, internalisation and immune response.**
(XLSX)

**S4 Table. Differentially expressed genes (DEG) in gill selected from enriched biological process and metabolic pathways related to non-self-recognition, internalisation and immune response.**
(XLSX)

## Acknowledgments

This study was under the academic responsibility of JMMS. JALC is the recipient of a doctoral fellowship (CONACYT-301921), under the academic direction of FAO and JMMS. They conceived and assessed the experimental design and research development and JALC performed zoo-technical work, collected and analysed data, and wrote the first draft of the manuscript. All authors revised and approved the final manuscript. The authors are grateful to the technical staff at CIBNOR: Delfino Barajas, Pablo Ormart, Julián Garzón, Carmen Rodríguez, Patricia Hinojosa, Norma Ochoa, Martín Ramírez, Martha Reyes and Eulalia Meza for their support.

Diana Fischer provided editorial services in English. Noroeste Sustentable A.C. donated scallops produced by Acuacultura Robles hatchery.

## Author Contributions

**Conceptualization:** José Manuel Mazón-Suástegui, Miguel Ángel Hernández-Oñate, Guadalupe Fabiola Arcos-Ortega.

**Data curation:** Jesús Antonio López-Carvallo.

**Formal analysis:** Jesús Antonio López-Carvallo, Miguel Ángel Hernández-Oñate.

**Funding acquisition:** José Manuel Mazón-Suástegui.

**Investigation:** José Manuel Mazón-Suástegui.

**Project administration:** José Manuel Mazón-Suástegui.

**Resources:** José Manuel Mazón-Suástegui.

**Software:** Miguel Ángel Hernández-Oñate.

**Supervision:** Miguel Ángel Hernández-Oñate, Guadalupe Fabiola Arcos-Ortega.

**Validation:** Jesús Antonio López-Carvallo, Dariel Tovar-Ramírez, Rosa María Morelos-Castro.

**Visualization:** Fernando Abasolo-Pacheco.

**Writing – original draft:** Jesús Antonio López-Carvallo.

**Writing – review & editing:** Jesús Antonio López-Carvallo, José Manuel Mazón-Suástegui, Miguel Ángel Hernández-Oñate, Guadalupe Fabiola Arcos-Ortega.

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
