## [Decision Letter · Decision Letter 0]

1 Apr 2020

PONE-D-20-00449

“TRANSCRIPTOME ANALYSIS OF CATARINA SCALLOP (Argopecten ventricosus) JUVENILES TREATED WITH HIGH-DILUTED IMMUNOMODULATORY COMPOUNDS REVEALS ACTIVATION OF NON-SELF-RECOGNITION SYSTEM”

PLOS ONE

Dear Dr. Guadalupe Fabiola Arcos,

Thank you for submitting your manuscript to PLOS ONE. After careful consideration, we feel that it has merit but does not fully meet PLOS ONE’s publication criteria as it currently stands. Therefore, we invite you to submit a revised version of the manuscript that addresses the points raised during the review process.

We would appreciate receiving your revised manuscript by May 16 2020 11:59PM. To enhance the reproducibility of your results, we recommend that if applicable you deposit your laboratory protocols in protocols.io, where a protocol can be assigned its own identifier (DOI) such that it can be cited independently in the future. For instructions see: http://journals.plos.org/plosone/s/submission-guidelines#loc-laboratory-protocols

We look forward to receiving your revised manuscript.

Kind regards,

Shawky M. Aboelhadid, PhD

Academic Editor

PLOS ONE

Reviewers' comments:

Reviewer's Responses to Questions

**Comments to the Author**

1. Is the manuscript technically sound, and do the data support the conclusions?

Reviewer #1: Yes

Reviewer #2: Yes

2. Has the statistical analysis been performed appropriately and rigorously? 

Reviewer #1: N/A

Reviewer #2: Yes

3. Have the authors made all data underlying the findings in their manuscript fully available?

Reviewer #1: Yes

Reviewer #2: Yes

4. Is the manuscript presented in an intelligible fashion and written in standard English?

Reviewer #1: Yes

Reviewer #2: Yes

5. Review Comments to the Author

Reviewer #1: López-Carvallo et al., try to demonstrate, through the study of the transcriptome, the interactions of juvenile A. ventricosus and highly-diluted treatments. Also is the first assembled transcriptome of the scallop A. ventricosus juveniles using RNA-seq technology, which allows us to generate biological markers.

Minor Comments

The microbiological point of view, the abbreviation of the species (spp) is not written in italics

The centrifugation units must be xg and not rpm

was vortexed at 3200…. Change was vortexed for

When handling animals, what are the rules of the country for euthany and go through an ethics committee to establish the number of specimens to use for such treatment

It is a good work, because they used the exactly methods for everything. The results and conclusions are supported by the methodology and discussion. The title reflects the finding of the manuscript.

The manuscript is for the most part well written and shows robust methods, though some grammatical corrections will be necessary and the use of prepositions should be reviewed. The abstract section is appropriate

Although the introduction is appropriate to the study purpose.

The figures must be improved

Reviewer #2: see my comments within the revised manuscript:

comment 1: Need reference in line 192.

comment 2: Which company, country you obtained marine broth.

comment 3: Line 206, Did you make sterility test for the HDIC solutions originated from the used vibrios??

comment 4:line 263: This title of this section may be changed to “Transcriptome, annotation and statistical analyses”

6. PLOS authors have the option to publish the peer review history of their article (what does this mean?). If published, this will include your full peer review and any attached files.

Reviewer #1: No

Reviewer #2: No

---

## [Author Response · Author response to Decision Letter 0]

16 Apr 2020

Dear Editor:

We want to thank you and the anonymous reviewers for their review and suggestions to our manuscript. We have carefully revised the manuscript according to reviewers’ comments and suggestions. We trust this manuscript has been improved and it is suitable for publication. The suggestions and comments were attended point-by-point as following:

Best regards

Changes suggested by reviwers´ are highlighted in the revised manuscript with tracks changes. 

Reviewer #1

Reviewer #1: López-Carvallo et al., try to demonstrate, through the study of the transcriptome, the interactions of juvenile A. ventricosus and highly-diluted treatments. Also is the first assembled transcriptome of the scallop A. ventricosus juveniles using RNA-seq technology, which allows us to generate biological markers.

Reviewer #1: Minor Comments

Reviewer #1: The microbiological point of view, the abbreviation of the species (spp) is not written in italics.

Answer:

NewLine 66 and 70: We removed italic format of “spp.”. 

Reviewer #1: The centrifugation units must be xg and not rpm.

Answer:

NewLine 195-196: The centrifugation units were changed to xg in accordance to reviewer#1 suggestions.

NewLine 201, 204 and 207: For agitation conditions, we decide to keep “rpm” for vortex speed as they are the units provided by manufacturer for the speed used (higher speed) (http://www.benchmarkscientific.com/BenchMixer.html).

Reviewer #1: was vortexed at 3200…. Change was vortexed for

Answer:

NewLine 199-201: “Vortexed at 3200...” was changed to “vortexed between each cycle for 2 min (3200 rpm)” in the Manuscript.

NewLine 203-204: “Vortexed at 3200...” was changed to “vortexed for 2 min (3200 rpm)” in the Manuscript.

Reviewer #1: When handling animals, what are the rules of the country for euthany and go through an ethics committee to establish the number of specimens to use for such treatment.

Answer:

Animal management was performed accordingly to ethics statements following the Official Mexican NORM (NOM-062-ZOO-1999), which considers the process of euthanasia as a humanitarian procedure used to end the life of laboratory animals, without causing pain or suffering. The Centro de Investigaciones Biológicas del Noroeste (CIBNOR), institution where the experiment was assessed, has an ethics committee that monitors the procedures with respect to the use of animals in the laboratory (Internal Committee for the Care and Use of Laboratory Animals, CICUAL). 

According to the CICUAL recommendations, the number of sampled organisms contemplated "the rule of maximizing information published and minimizing unnecessary studies". In this sense, for this work, we considered the minimum number of organisms needed to obtain a high-quality transcriptome.

NewLine 883-897: 

Ethics statement

All organisms used in this work were handled in accordance with the Official Mexican Standard protocols (NOM-062-ZOO-1999). Argopecten ventricosus juveniles were provided from the non-governmental association Noroeste Sustentable (NOS) and transported to aquarium tanks at CIBNOR with all the required permits from the federal agency CONAPESCA. All animals used in this study were produced by Acuacultura Robles hatchery in captivity for experimental purposes. The organisms were kept in optimal culture conditions to avoid stressful conditions and no harmful effects have been detected using HDIC (Highly-Diluted Immunomodulatory Compounds) in marine organisms.

Reviewer #1: It is a good work, because they used the exactly methods for everything. The results and conclusions are supported by the methodology and discussion. The title reflects the finding of the manuscript. The manuscript is for the most part well written and shows robust methods, though some grammatical corrections will be necessary and the use of prepositions should be reviewed. The abstract section is appropriate

Although the introduction is appropriate to the study purpose.

Answer:

M.S. Diana Fischer (Official translator-SGA-294/2007) provided editorial services of overall grammar and general English to improve the readability of the paper as recommended by the reviewer#1. Changes can be found in the document “Revised Manuscript with Track Changes.doc” 

Reviewer #1: The figures must be improved

Answer:

All figures were uploaded to the Preflight Analysis and Conversion Engine (PACE) digital diagnostic tool to ensure they met PLOS requirements, as suggested by Reviewer#1. 

Reviewer #2

Reviewer #2: see my comments within the revised manuscript:

Reviewer #2, Line177: Delete posterior

Answer:

NewLine 179: The word “posterior” was deleted.

Reviewer #2, Line191-192: Need reference in line 192

Answer:

NewLine 194: The reference “[34]” was included. As the reference [34] was added to the manuscript, the number of the affected references was changed.

Reviewer #2, Line194: Which company, country you obtained marine broth.

Answer:

NewLine 196-197: More details to describe from which company and country we obtained the marine broth was added as: “of 2 L of Vibrio culture in marine broth 2216 (105 × 10-6 CFU mL-1)” to “of 2 L of Vibrio culture (105 × 10-6 CFU mL-1) in marine broth 2216 (BD DifcoTM, USA)”. 

Reviewer #2, Line198: Change “two” for “2”

Answer:

NewLine201: The word “two” was changed for “2”.

Reviewer #2, Line201: Change “two” for “2”

Answer:

NewLine204: The word “two” was changed for “2”.

Reviewer #2, Line204: Change “two” for “2”

Answer:

NewLine207: The word “two” was changed for “2”.

Reviewer #2, Line 206, Did you make sterility test for the HDIC solutions originated from the used vibrios??

Answer:

Biomass for the preparation of Vibrios lysates was inactivated by freezing-unfreezing cycles (-80oC), followed by a mechanical disruption using vortex for 2 min (3200 rpm, between each cycle) and the subsequent addition of ethanol. All HDIC solutions were prepared under sterile conditions.

Reviewer #2, Line 263: This title of this section may be changed to “Transcriptome, annotation and statistical analyses” 

Answer:

NewLine267-268: “Transcriptome analysis and annotation” was changed to “Transcriptome, annotation and statistical analyses”.

---

## [Decision Letter · Decision Letter 1]

28 Apr 2020

“TRANSCRIPTOME ANALYSIS OF CATARINA SCALLOP (Argopecten ventricosus) JUVENILES TREATED WITH HIGH-DILUTED IMMUNOMODULATORY COMPOUNDS REVEALS ACTIVATION OF NON-SELF-RECOGNITION SYSTEM”

PONE-D-20-00449R1

Dear Dr. Guadalupe Fabiola Arcos,

We are pleased to inform you that your manuscript has been judged scientifically suitable for publication and will be formally accepted for publication once it complies with all outstanding technical requirements.

With kind regards,

Shawky M. Aboelhadid, PhD

Academic Editor

PLOS ONE

Additional Editor Comments (optional):

Reviewers' comments:

Reviewer's Responses to Questions

**Comments to the Author**

1. If the authors have adequately addressed your comments raised in a previous round of review and you feel that this manuscript is now acceptable for publication, you may indicate that here to bypass the “Comments to the Author” section, enter your conflict of interest statement in the “Confidential to Editor” section, and submit your "Accept" recommendation.

Reviewer #1: All comments have been addressed

Reviewer #2: (No Response)

2. Is the manuscript technically sound, and do the data support the conclusions?

Reviewer #1: Yes

Reviewer #2: Yes

3. Has the statistical analysis been performed appropriately and rigorously? 

Reviewer #1: Yes

Reviewer #2: Yes

4. Have the authors made all data underlying the findings in their manuscript fully available?

Reviewer #1: Yes

Reviewer #2: Yes

5. Is the manuscript presented in an intelligible fashion and written in standard English?

Reviewer #1: Yes

Reviewer #2: Yes

6. Review Comments to the Author

Reviewer #1: The authors responded point by point all comments so it should be accepted.

It is a good work, because they used the exactly methods for everything. The results and conclusions are supported by the methodology and discussion. The title reflects the finding of the manuscript.

Reviewer #2: (No Response)

7. PLOS authors have the option to publish the peer review history of their article (what does this mean?). If published, this will include your full peer review and any attached files.

Reviewer #1: No

Reviewer #2: No

---

## [Editor Report · Acceptance letter]

1 May 2020

PONE-D-20-00449R1 

Transcriptome analysis of Catarina scallop (*Argopecten ventricosus*) juveniles treated with highly-diluted immunomodulatory compounds reveals activation of non-self-recognition system 

Dear Dr. Arcos-Ortega:

I am pleased to inform you that your manuscript has been deemed suitable for publication in PLOS ONE. Congratulations! Your manuscript is now with our production department. 

With kind regards,

on behalf of

Professor Shawky M. Aboelhadid 

Academic Editor

PLOS ONE